# Assembly assay identifies a critical region of human fibrillin-1 required for 10–12 nm diameter microfibril biogenesis

Sacha A. Jensen[¤a]*, Ondine Atwa[¤b], Penny A. Handford*

Department of Biochemistry, University of Oxford, Oxford, United Kingdom

¤a Current address: College of Public Health, Medical and Veterinary Sciences, James Cook University, Townsville, Queensland, Australia
¤b Current address: Cardiovascular Disease Initiative & Precision Cardiology Lab, Broad Institute, Cambridge, Massachusetts, United States of America
* sacha.jensen@jcu.edu.au (SAJ); penny.handford@bioch.ox.ac.uk (PAH)

**Data Availability Statement:** All relevant data are within the paper and its Supporting information files.

**Funding:** SAJ and PAH gratefully acknowledge support from Arthritis Research UK

## Abstract

The human *FBN1* gene encodes fibrillin-1 (FBN1); the main component of the 10–12 nm diameter extracellular matrix microfibrils. Marfan syndrome (MFS) is a common inherited connective tissue disorder, caused by *FBN1* mutations. It features a wide spectrum of disease severity, from mild cases to the lethal neonatal form (nMFS), that is yet to be explained at the molecular level. Mutations associated with nMFS generally affect a region of FBN1 between domains TB3-cbEGF18—the "neonatal region". To gain insight into the process of fibril assembly and increase our understanding of the mechanisms determining disease severity in MFS, we compared the secretion and assembly properties of FBN1 variants containing nMFS-associated substitutions with variants associated with milder, classical MFS (cMFS). In the majority of cases, both nMFS- and cMFS-associated neonatal region variants were secreted at levels comparable to wild type. Microfibril incorporation by the nMFS variants was greatly reduced or absent compared to the cMFS forms, however, suggesting that nMFS substitutions disrupt a previously undefined site of microfibril assembly. Additional analysis of a domain deletion variant caused by exon skipping also indicates that register in the neonatal region is likely to be critical for assembly. These data demonstrate for the first time new requirements for microfibril biogenesis and identify at least two distinct molecular mechanisms associated with disease substitutions in the TB3-cbEGF18 region; incorporation of mutant FBN1 into microfibrils changing their integral properties (cMFS) or the blocking of wild type FBN1 assembly by mutant molecules that prevents late-stage lateral assembly (nMFS).

## Introduction

Fibrillins are large (~350 kDa) multifunctional glycoproteins that constitute the major structural component of the 10–12 nm diameter microfibrils of the extracellular matrix (ECM) in metazoan species [1]. In addition to providing a scaffold for the deposition of elastin during

(www.versusarthritis.org; project grant no. 20785) and the MRC (https://mrc.ukri.org; grant number MR/M009831/1).

**Competing interests:** The authors have declared that no competing interests exist.

elastic fibre synthesis in higher vertebrates, microfibrils play a role in tissue homeostasis and development through their interactions with growth factors such as transforming growth factor β (TGFβ) and the bone morphogenetic proteins (BMP). They also influence cell-matrix communication via interactions with the integrins $\alpha_8\beta_1$, $\alpha_v\beta_1$, $\alpha_v\beta_5$, $\alpha_v\beta_3$, $\alpha_v\beta_6$, $\alpha_{IIb}\beta_3$ and $\alpha_5\beta_1$ [2–7]. In humans, three fibrillin isoforms, FBN1, -2 and -3, are expressed from separate genes [8–12]. The majority of mutations affecting FBN1 result in Marfan syndrome (MFS); one of the most common human connective tissue disorders affecting the cardiovascular, skeletal and ocular systems.

A characteristic feature of diseases caused by mutations in *FBN1* is their marked phenotypic variability, with differences in disease severity being seen even between members of a single family carrying the same disease allele in cases of MFS [13]. Few clear genotype-phenotype correlations have been established except in cases of rare diseases caused by mutations in specific FBN1 domains. Stiff skin syndrome, for example, is caused by mutations affecting domain TB4 [14] while the dominant forms of geleophysic dysplasia and acromicric dysplasia are linked to substitutions in domain TB5 [15]. In both of these cases, the mutant proteins secrete and incorporate into matrix [16]. In contrast, mutations affecting domains TB4 and TB5 that lead to intracellular retention of the mutant protein cause MFS [16], consistent with a functional haploinsufficiency mechanism in these cases. Genotype-phenotype correlations that explain MFS severity are rarer although an exception is neonatal MFS (nMFS), which is linked to a cluster of mutations affecting domains TB3-cbEGF18 of FBN1; the "neonatal region". Cases of nMFS are atypically severe, presenting with symptoms of MFS at birth, and are generally lethal in infancy. At present, little is known about the molecular mechanisms underlying nMFS although previous work has shown that FBN1 nMFS variants show altered protease susceptibilities and a reduced affinity for heparin [17].

When viewed by rotary shadowing, isolated fibrillin microfibrils have a beaded filament appearance with an average untensioned periodicity of 50–60 nm [18–20]. The domain organisation of the fibrillins has been highly conserved throughout evolution [21–23], being dominated by 43 calcium-binding epidermal growth factor-like (cbEGF) domains interspersed with 7 transforming growth factor β-binding protein-like (TB) domains (Fig 1A). Although the organisation of fibrillin monomers within the microfibrils is still a matter of controversy [24, 25], a variety of molecular and cellular data have suggested a model of microfibril assembly dependent on pericellular proteolysis, heparan sulphate interactions and regulated multimerisation. At the point of secretion, or immediately afterwards, fibrillin is processed by furin, which removes a C-terminal propeptide that regulates assembly [26, 27]. The exposed C-terminal domains then associate into bead-like structures [28], enhancing the apparent affinity of the region for the N-terminal domains of other fibrillin molecules and promoting the head-to-tail alignment of the polypeptides. Throughout the process, avidity-driven interactions with cell surface heparan sulphate proteoglycans limit the diffusion of the nascent microfibrils [29–31].

To investigate the pathogenic mechanisms leading to the wide spectrum of disease severity seen in MFS, we compared the fibroblast secretion profiles of a series of FBN1 mini-gene constructs harbouring nMFS or cMFS mutations within the neonatal region. The secretion profiles of full-length, GFP-tagged FBN1 nMFS and cMFS variants were then compared using a HEK293T-based transient transfection assay. In most cases, the mutant proteins secreted at levels similar to wild type. When assessing the incorporation of the GFP-tagged FBN1 variants into the ECM through co-cultures with fibroblasts, however, clear differences were seen between the two classes with incorporation of the nMFS mutants being greatly reduced or absent compared to the cMFS forms. Our results provide further evidence of a dominant negative mechanism in cases of cMFS involving cbEGF domain substitutions and show that nMFS

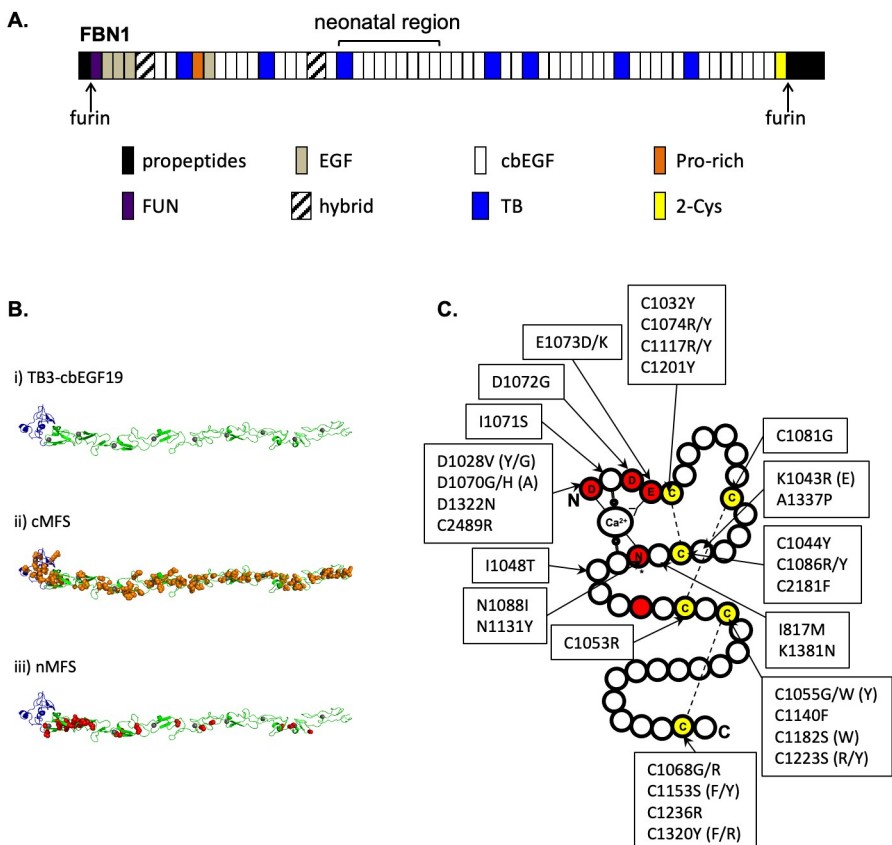

**Fig 1. FBN1 domain organisation and sites of neonatal Marfan syndrome mutations. A)** The structure of FBN1 is dominated by cbEGF domains interspersed with TB domains. The majority of nMFS-associated mutations affect the central TB3- cbEGF18 domains, which have traditionally been referred to as the "neonatal region". **B)** A model of the neonatal region (i) was created using Modeller software [35] and coordinates from the structures of domains TB4 [4], cbEGF12-13 [36] and cbEGF32-33 [37]. TB3 is coloured blue, cbEGFs are coloured green with $Ca^{2+}$ ions shown as grey spheres. Substitutions associated with cMFS (panel ii, orange) are found evenly distributed across the neonatal region while those associated with nMFS (panel iii, red) show a clustering around cbEGF11-12. **C)** Plotting the positions of known nMFS-associated substitutions on a schematic representation of a cbEGF domain shows that in many cases the effect of the substitution is structural, affecting either cysteines involved in disulphide bond formation (yellow) or $Ca^{2+}$-binding consensus residues (red). Where nMFS and cMFS substitutions have been found at the same position, the cMFS version is shown in brackets. Data extracted from the Universal Mutation Database (http://umd.be/FBN1) [38].

results from the disruption of a microfibril assembly site (or sites) within domains TB3-cbEGF18, with mutant molecules likely blocking the late-stage assembly of wild-type FBN1. These data also show that the length and $Ca^{2+}$-dependent structure of the TB3-cbEGF18 region are both critical for native assembly of the 10–12 nm microfibril.

# Materials and methods

## Plasmid construction, mutagenesis and transfection

Construction of the plasmid pcDNA-GFPFBN has been described previously [26]. Plasmids for the expression of the NterPro-TB3cbEGF19 constructs were created by amplifying the DNA encompassing encoding residues D952 to T1403 as a *Sal*I fragment from a *FBN1* cDNA clone and inserting this into the *Xho*I site of the plasmid pKG52(polyA) [32], which encodes the N-terminal region of FBN1 up to the proline-rich domain (residues 1–446) under the

control of a thymidine kinase promoter. Mutant constructs were created by overlap extension PCR [33].

For the transfection of MSU-1.1 fibroblasts, 2.5 μg of plasmid DNA was mixed with 5 μl TranIT-X2 transfection reagent (Mirus) in 250 μl OPTIMEM reduced serum medium and incubated at room temperature for 20 minutes. DNA complexes were then added to cells at ~95% confluence in a well of a 6-well tissue culture dish with 2.5 ml Dulbecco's modified Eagle medium supplemented with 2 mM glutamine, 50 U/ml penicillin, 50 μg/ml streptomycin and 10% (v/v) fetal bovine serum (complete DMEM). After 24 hours, cells were harvested with trypsin and transferred to 75 cm$^2$ tissue culture flasks. Selection was commenced 4 days after transfection with 2.5 μg/ml puromycin in complete DMEM, with medium changes every 2–3 days for 3 weeks. Pooling of clones (typically >80 clones per pool) was used to average out the effects of variations in expression levels between individual clones resulting from random genomic integration and to overcome the low transient transfection efficiency of fibroblasts.

Transfections of HEK293T cells with GFP-FBN1 constructs were carried out in 6-well plates using 3 μl of TransIT-X2 transfection combined with 500 ng DNA in 100 μl OPTIMEM reduced serum medium. After 20 minutes incubation, transfection complexes were added to cells in 2.5 ml complete DMEM and incubated overnight before harvesting for either secretion or microfibril incorporation assays.

## Microfibril incorporation assay

Microfibril incorporation assays were carried out as described previously [16, 26] with some modifications. Briefly, HEK293T cells were transfected in 6-well plates overnight then trypsinised and counted. Co-cultures on 13 mm diameter glass coverslips were established using 2 x 10$^5$ FS2 dermal fibroblasts [14] and 2 x 10$^5$ transfected HEK293T cells per well of 24 well plates. Cells were cultured in complete DMEM for five days then fixed with 4% (w/v) paraformaldehyde in phosphate-buffered saline and stained using a rabbit polyclonal antibody raised against the FBN1 proline-rich region [32] and chicken polyclonal anti-GFP (Abcam; ab13970) without permeabilisation. Goat anti-chicken Alexa488 and goat anti-rabbit Alexa568 (Invitrogen) were used for detection. Images were collected using a Zeiss Axioplan 2 microscope with AxioVision Rel. 4.8 software.

## Immunoblotting

Cells remaining from the HEK293T transfections were transferred to 25 cm$^2$ tissue culture flasks (Greiner) and grown for a further 3 days in complete DMEM to produce cell and medium samples for analysis by Western blotting. Conditioned media and cell samples were analysed by immunoblotting as described previously [32, 34] using a chicken anti-GFP antibody followed by a goat anti-chicken HRP conjugate and enhanced chemiluminescent detection (Amersham).

## Results

### Neonatal MFS sites are unevenly distributed across FBN1 domains TB3-cbEGF18

FBN1 domains TB3-cbEGF18 (Fig 1A) have traditionally been referred to as the "neonatal region" since the vast majority of mutations identified in cases of nMFS cluster in these domains. To gain an insight into the different molecular mechanisms underlying nMFS and cMFS, we used the Universal Mutation Database entry for FBN1 (http://www.umd.be/FBN1/) to map the positions of known nMFS-associated mutations within *FBN1* (Fig 1B; S1 Table). As

expected, the majority of nMFS-associated mutations affected domains TB3-cbEGF18. Mutations affecting regions outside this area are rarer and include 1 bp or 2 bp deletions leading to frameshifts (c.390delT and c.7167_7168delCT), deletions in the cDNA spanning multiple domains (c.2294_2854del, c.5423_5788del and c.5546_5917del corresponding to ΔcbEGF8-10, ΔcbEGF26-28 and ΔcbEGF27-29, respectively), splice variants (c.IVS35-32del20a) and substitutions (C129Y, I817M, C2181F, C2489R). Interestingly, the distribution of nMFS-associated substitutions within domains TB3-cbEGF18 is uneven, with half of the mutations affecting only domains cbEGF11 and cbEGF12 (Fig 1B). In contrast, cMFS-associated substitutions within domains TB3-cbEGF18 (S2 Table) are more evenly distributed, suggesting that mutations affecting domains cbEGF11 and cbEGF12 are particularly likely to result in nMFS. Additionally, there are cases in which different substitutions at the same amino acid position lead to either cMFS or nMFS (Fig 1C), suggesting that both position and residue type are important in determining disease severity.

Looking more closely at the types of nMFS-associated substitutions that occur in cbEGF domains, most (33/38) are predicted to have structural consequences that would alter the fold of the domain, affecting residues involved in either calcium-binding or disulphide bond formation (Fig 1C). Of the cysteine substitutions, the majority reported in the UMD (20/22) affect either the Cys1-Cys3 or Cys5-Cys6 disulphide, with only two affecting the Cys2-Cys4 disulphide (C1081G and C1053R). Of the remaining five substitutions, I1048T has previously been reported to introduce an N-linked glycosylation site [39] and also affects a predicted site of interdomain packing between domains TB3 and cbEGF11 [33]. Both of these effects would have structural consequences. The remaining four substitutions (I817M, K1043R, A1337P, K1381N) are not obviously structural, but could indicate patches involved in intermolecular interactions.

## Secretion from fibroblasts of MFS-associated FBN1 neonatal region mutants

To compare the effects of substitutions causing nMFS or cMFS on FBN1 secretion from fibroblasts, we used a previously described assay [32, 34] in which pools of stably transfected clones of the human fibroblast cell line, MSU-1.1, are created to express a FBN1 mini-gene construct (Fig 2A). MSU-1.1 fibroblasts are able to assemble extracellular microfibrils and express all the cellular factors required for FBN1 folding, processing and secretion [40]. For this assay, the mini-gene construct encodes the N-terminal domains of FBN1 up to the proline-rich region fused to domains TB3-cbEGF19, spanning the entire neonatal region. The proline-rich region provides an epitope for the detection of the fusion construct (MW ~130 kDa) by immunoblotting. Five substitutions (I1048T, E1073K, C1086Y, C1111R, N1131Y) [41–48] and a domain deletion resulting from a splice site mutation (ΔcbEGF16) [49, 50] linked to nMFS were generated for comparison with a series of substitutions associated with milder disease (Fig 2B); either cMFS (Y1101C, D1113G, D1115G, N1382S) or isolated aortic aneurysm and dissection (G1127S) [42, 44, 49, 51–56].

In many cases, both nMFS and cMFS constructs were found to secrete from fibroblasts (Fig 2B). This was unexpected for the N1131Y, D1113G and D1115G variants in cbEGF13, which in a previous study were shown to be retained in cells [34]. The I1048T variant migrated at a higher molecular weight compared to the wild-type construct, consistent with the introduction by this substitution of an additional N-linked glycosylation site [39]. The ΔcbEGF16 deletion construct migrated more rapidly as expected due to the loss of domain cbEGF16. Its presence in the medium indicates that any structural changes to the molecule resulting from the new

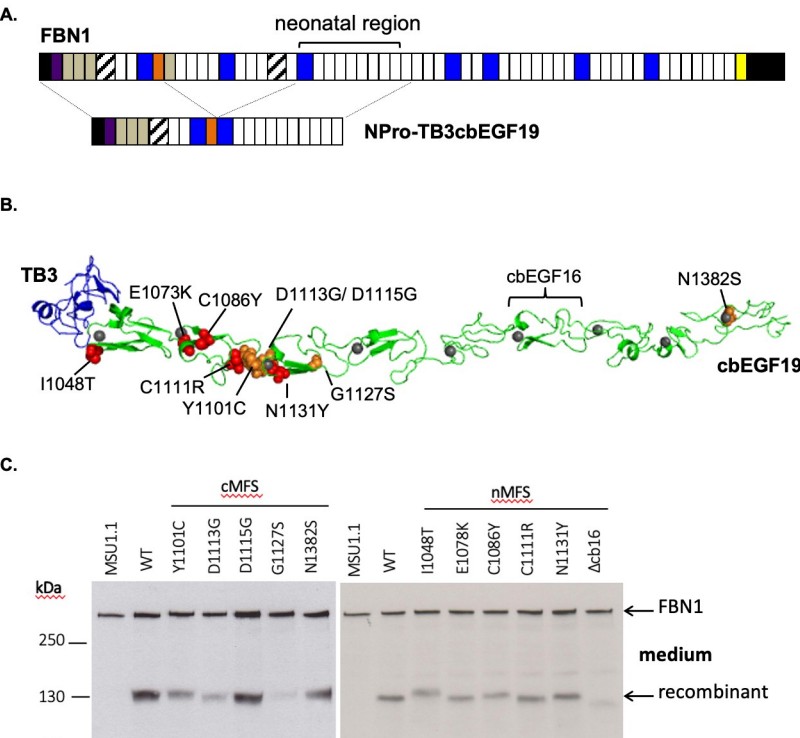

**Fig 2. Construction of the NterPro-TB3cbEGF19 mini-gene and secretion of cMFS and nMFS variants from fibroblasts. A)** NterPro-TB3cbEGF19 is a fusion of the N-terminal domains of FBN1 up to the proline-rich region (orange) with domains TB3 to cbEGF19, which encompass the neonatal region. Western blotting using an antibody directed against the proline-rich domain (orange) is used to distinguish the fusion construct from endogenous FBN1 expressed by fibroblasts on the basis of size. **B)** Model of the TB3-cbEGF19 domains highlighting the positions of the nMFS (red spheres) and cMFS (orange spheres) substitutions described in this work, and the position of domain cbEGF16, which is deleted in one of the nMFS variants. Domain TB3 is coloured blue, cbEGF domains are coloured green and $Ca^{2+}$ ions bound to the cbEGF domains are shown as grey spheres. **C)** Secretion profiles of the neonatal region mutants associated with nMFS and cMFS. Medium samples from untransfected MSU-1.1 fibroblasts (MSU-1.1) and fibroblasts transfected with the wild-type (WT) construct were used as controls to allow the identification of the recombinant construct (MW ~130kDa, arrowed). Full-length FBN1, expressed endogenously by the fibroblasts, functions as a loading control (MW ~350kDa, arrowed).

interdomain packing interaction between domains cbEGF15 and cbEGF17 do not block secretion.

## nMFS and cMFS mutants differ in their capacity to incorporate into microfibrils

To assess the ability of nMFS and cMFS variants to incorporate into microfibrils in the context of a full-length FBN1 molecule, we used a microfibril incorporation assay [26] in which HEK293T cells, transiently transfected with wild-type or mutant versions of GFP-tagged FBN1 (GFP-FBN1), are co-cultured with human dermal fibroblasts. Human dermal fibroblasts are difficult to transfect and the production of stably transfected lines expressing a construct of the size of full-length FBN1 (coding sequence of 8.6 kb) is extremely inefficient. Transient trans-fection of HEK293T cells in this assay overcomes these difficulties. HEK293T cells transfect with high efficiency, but are unable to assemble a microfibril network, so the ability of tagged mutant molecules to assemble in co-cultures with fibroblasts can be assessed clearly.

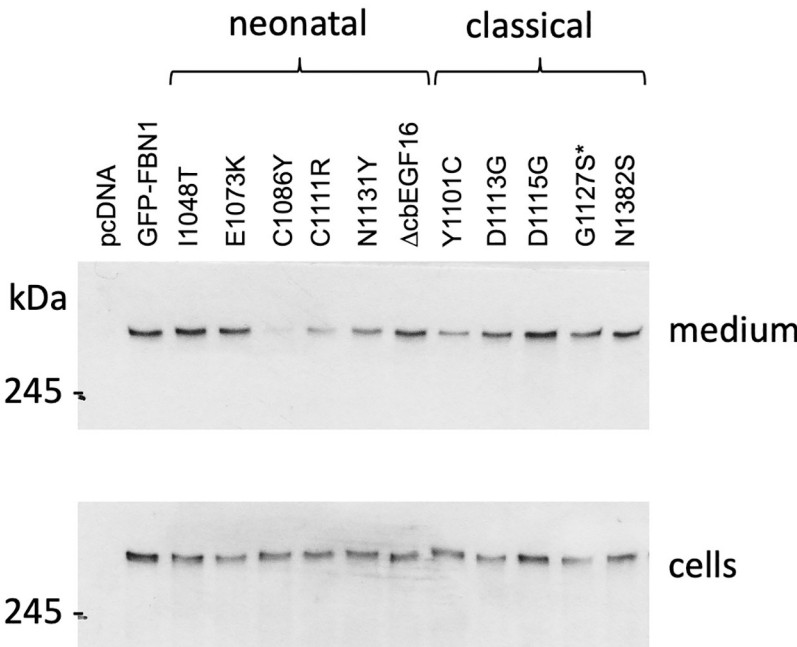

**Fig 3. Secretion of full length GFP-FBN1 fusion constructs from HEK293T cells.** Mutations associated with nMFS (neonatal) or cMFS (classical) were engineered into GFP-tagged FBN1 cDNA construct and transiently transfected into HEK293T cells. After 3 days of culture, samples of cells and medium fraction were analysed by Western blotting with an antibody directed against the GFP epitope. Empty vector (pcDNA) and wild-type construct (GFP-FBN1) were used as negative and positive controls. The majority of mutant constructs were easily detected in the culture medium. The C1086Y variant was consistently seen at reduced levels in the medium compared to the other constructs. Cell fraction samples indicated that expression levels were similar across the different variants and that the reduction seen for the C1086Y variant was not due to a lack of expression. *The G1127S substitution, which was identified in cases of isolated aortic aneurysm rather than cMFS, is associated with a milder form of disease and so is grouped with the cMFS variants here.

When expressed in HEK293T, full-length GFP-FBN1 fusion constructs behaved similarly to the mini-gene constructs used to assess secretion from fibroblasts. In most cases, the GFP fusions were detected on western blots of medium samples prepared from transiently transfected HEK293T cultures (Fig 3). The secretion of the nMFS-associated C1086Y variant was found to be greatly reduced compared to wild-type GFP-FBN1 although it was still detectable on western blots, in contrast to previously studied cysteine—to—tyrosine substitutions affecting FBN1 domains TB4 and TB5 [16]. Similarly, the nMFS-associated C1111R variant showed reduced but readily detectable levels of secretion from HEK293T compared to wild-type GFP-FBN1. Little difference was observed in the apparent migration of the I1048T and ΔcbEGF16 constructs compared to wild-type in these experiments, in contrast to the fibroblast secretion data, as a result of the high molecular weight of the GFP-FBN1 fusion and the low resolving power of SDS-PAGE in this size range.

To assess the incorporation of the nMFS and cMFS mutants into microfibrils, co-cultures of FS2 human dermal fibroblasts and transiently transfected HEK293T cells were grown for 5 days at confluence, fixed and stained for GFP and FBN1 without permeabilisation (Figs 4 and 5). Omitting a permeabilisation step makes it possible to distinguish between the GFP-reactive material outside cells and the GFP-fusion inside the HEK293T cells, which will not be stained by the anti-FBN1 antibody and appears as green "spots" in the merged immunofluorescence images. When co-cultured with FS2 fibroblasts, clear differences were seen in the behaviour of the GFP-FBN1 constructs harbouring nMFS versus cMFS mutations. Very little or no evidence

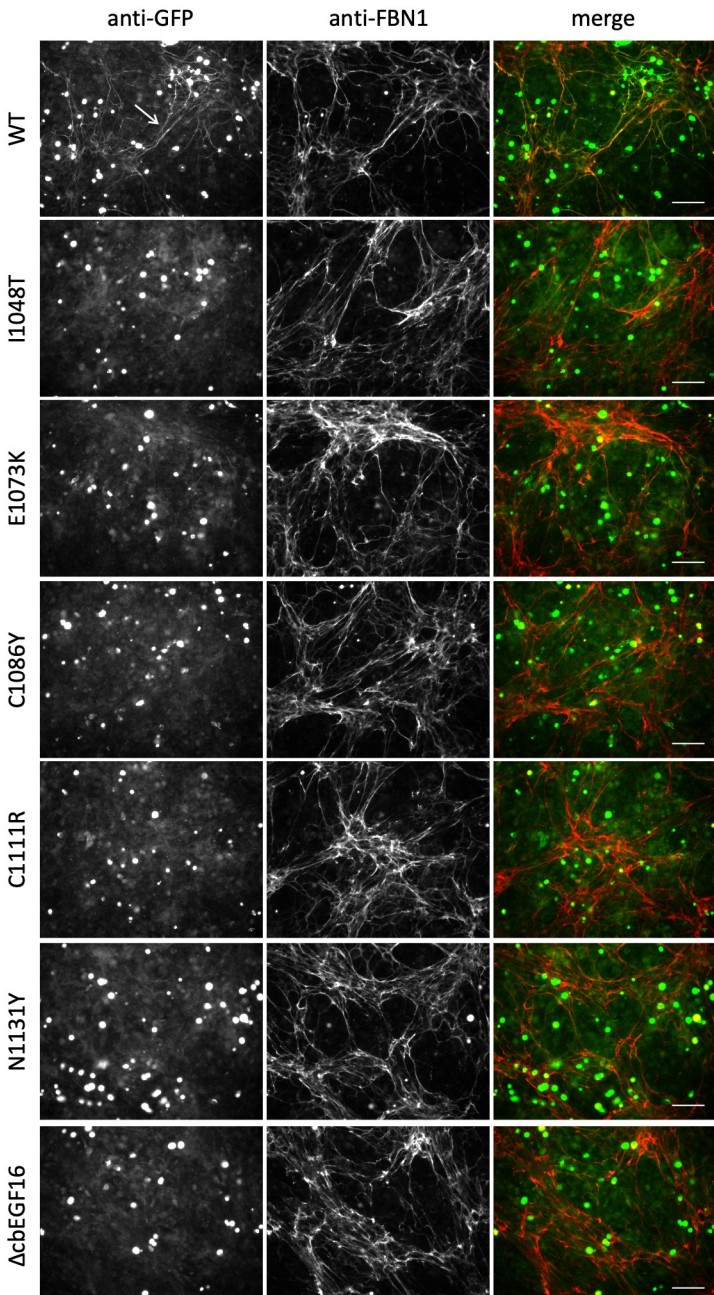

**Fig 4. Microfibril incorporation of GFP-FBN1 nMFS variants.** FS2 fibroblasts were co-cultured for 5 days with HEK293T cells transiently transfected to express GFP-FBN1 (WT) or variants associated with nMFS. Co-cultures were then fixed and stained with anti-GFP and anti-FBN1 antibodies without permeabilisation [16, 26]. Although the nMFS variants I1048T, E1073K, C1111R, N1131Y and ΔcbEGF16 were clearly detectable in medium samples when expressed as GFP-FBN1 fusions, microfibril networks labelled with the recombinant GFP-FBN1 (arrow in WT panel) were rarely seen. Bar = 100 μm.

of microfibril incorporation of the GFP fusion was seen in co-cultures of the nMFS mutants, in spite of their secretion from the HEK293T cells (Fig 4). Only rarely were fine strands of GFP-reactive material observed to co-localise with FBN1 (not shown). In contrast, the cMFS variants and the G1127S variant associated with an isolated aortic disease phenotype were

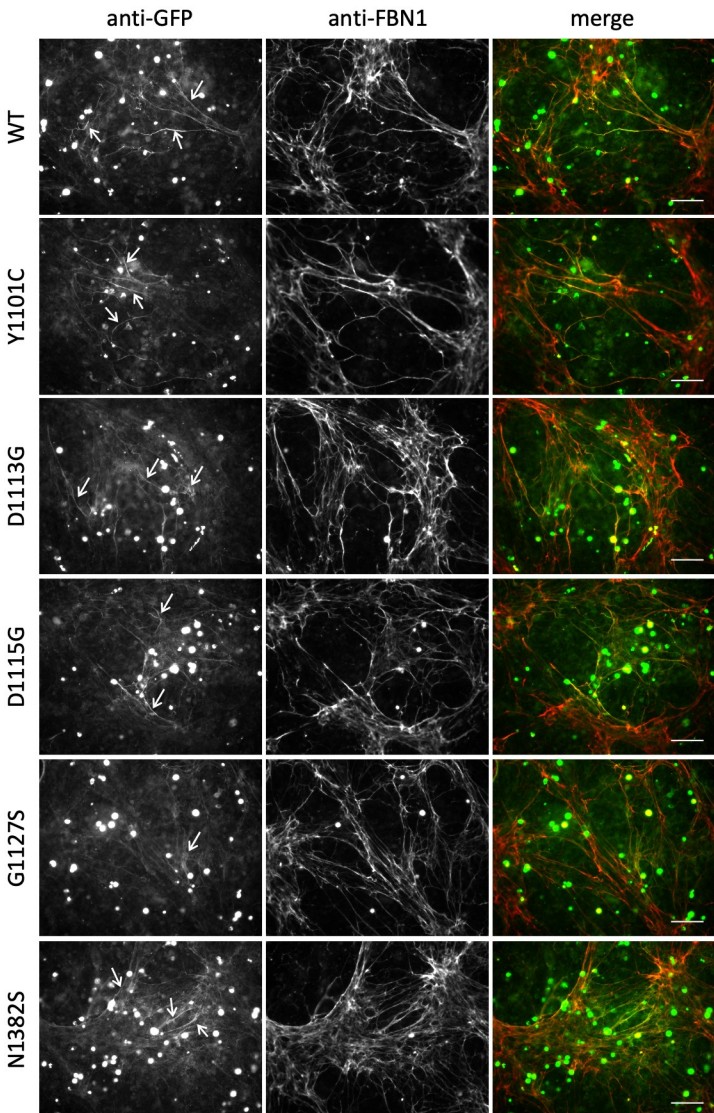

**Fig 5. Microfibril incorporation of GFP-FBN1 cMFS variants.** FS2 fibroblasts were co-cultured for 5 days with HEK293T cells transiently transfected to express GFP-FBN1 (WT) or variants associated with cMFS. Co-cultures were then fixed and stained with anti-GFP and anti-FBN1 antibodies without permeabilisation [16, 26]. In contrast to the nMFS variants, co-cultures expressing GFP-cMFS variants produced microfibril networks containing readily detectable recombinant material (white arrows). Bar = 100 μm.

easily detectable although the staining for these constructs in the matrix often appeared weaker than for wild-type GFP-FBN1 (Fig 5). These results show for the first time that the nMFS and cMFS mutants can be separated into two distinct classes based on their ability to incorporate into microfibrils using this assay, with the nMFS variants appearing to disrupt a motif(s) in FBN1 required for its incorporation into microfibrils.

## Discussion

The work presented here compares the molecular and cellular fates of FBN1 variants within the TB3-cbEGF18 region associated with classical and severe "neonatal" forms of MFS to gain an insight into the mechanisms that determine disease severity. Fundamental to the

development of MFS is the reduction in quantity and/or quality of FBN1-containing microfibrils in the ECM. FBN1 null alleles resulting from genomic deletions of the *FBN1* gene [57] have been reported and demonstrate the importance of haploinsufficiency in MFS pathogenesis. Early studies on the mgR and mgΔ mouse models also indicated that a critical threshold of microfibrils was necessary for the maintenance of elastic tissues such as the aorta [58, 59]. However, no satisfactory explanation for the difference in severity of disease in MFS has been forthcoming. Our results, utilising a microfibril incorporation assay with GFP-FBN1 disease-causing variants, together with fibroblast secretion analysis, now provide new evidence for two different mechanisms of microfibril dysfunction underlying cMFS and nMFS. Most variants analysed were secreted into the media of co-cultures, and showed no evidence of degradation, but only cMFS FBN1 variants incorporated readily into microfibrils. The nMFS variants, in contrast, appear to be blocked from incorporating into microfibrils, strongly suggesting the presence of a previously undefined assembly site within domains TB3-cbEGF18.

Our data suggest a plausible model, based on two different mechanisms, to explain the development of cMFS and nMFS associated with substitutions in the TB3-cbEGF18 region. In the milder cMFS form, mutant FBN1 molecules are able to assemble with wild type, or close to wild type, efficiency into microfibrils, as all necessary sites of assembly are present and functional. Although there may be differences in the incorporation efficiencies of the cMFS variants, this may not be detectable using the current assay system given its semi-quantitative nature. Incorporation of the cMFS-associated substitution C1663R in cbEGF24 [49, 60] has previously been reported in a transgenic mouse model [61]. Loss of microfibril function in these cases may develop gradually as a result of differences in the biomechanical properties of the mutant microfibrils compared to wild type, or susceptibility to degradation. In contrast, the nMFS mutants show greatly reduced or no incorporation into microfibrils indicating disruption of a previously unrecognised assembly site. In cases of nMFS, mutant and wild type molecules would be able to associate in the early assembly stages since they have native N-and C-termini capable of multimerisation (Fig 6A), however the nascent microfibrils would be blocked from further assembly due to defective heparin- or protein-protein interactions involving the neonatal region. This interaction would result in non-productive complexes (Fig 6B) that effectively reduce the pool of assembly-competent FBN1 to levels lower than those that would be expected for haploinsufficiency, leading to the more severe phenotype observed in nMFS.

Of the disease-causing substitutions investigated here, almost all affect domains cbEGF11-13 and are predicted to cause structural changes through effects on $Ca^{2+}$ binding (E1073K, D1113G, D1115G, N1131Y and N1328S), disulphide bond formation (C1086Y and C1111R), disruption of an interdomain packing site (I1048T and Y1101C), altered glycosylation (I1048T) or deletion of an entire domain (ΔcbEGF16). The majority were processed and secreted from cells at levels comparable to wild type. Surprisingly, only a subset including Y1101C (cbEGF12), D1113G, D1115G, and G1127S (cbEGF13), and N1382S (cbEGF19), incorporated into microfibrils, whilst I1048T (cbEGF11), E1073K, C1086Y, and C1111R (cbEGF12), N1131Y (cbEGF13), and ΔcbEGF16 did not. These data indicate that the structural changes have a different impact on assembly, which is not simply explained by levels of secretion. The exceptions to this may be C1086Y and C1111R, which showed a reduced level in the extracellular medium compared with other variants. The variable consequences for assembly and disease severity shown by the cbEGF13 calcium-binding substitutions D1113G, D1115G (cMFS) and N1131Y (nMFS) suggest a more significant functional role for N1131. This could be due to the degree of structural perturbation caused by loss of calcium binding to cbEGF13 and/or additional roles for the central β-hairpin on which residue N1131 is located. In EGF12 of the Notch receptor, residues on the central β-hairpin contribute to a specific binding

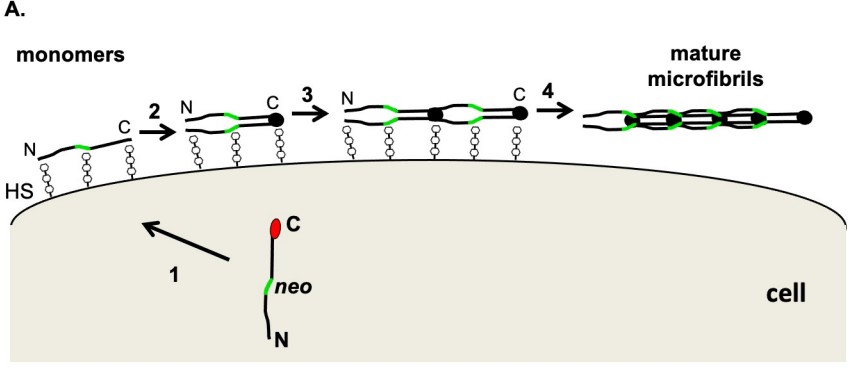

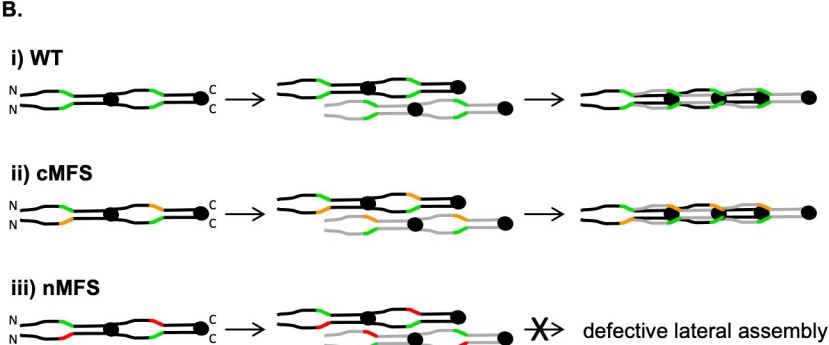

**Fig 6. Model of the FBN1 neonatal region's roles in microfibril assembly and nMFS pathogenesis. A**) In normal FBN1 assembly, secretion of the protein is coupled to the proteolytic cleavage (**1**) of the C-terminal propeptide (red circle). FBN1 interactions with heparan sulphate proteoglycans (HS), at sites including the N- and C-termini and neonatal region (other sites not shown), limit diffusion from the cell surface during assembly. Multimerisation of C-terminal domains, previously blocked by the propeptide, initiate the assembly process (**2**) and promote intermolecular avidity-driven interactions with N-terminal domains (**3**) that result in a head-to-tail alignment of monomers. The neonatal region (green) may be involved in the higher order, lateral assembly of FBN1 to form the mature microfibril (**4**). **B**) Lateral assembly (panel A, step 4) would occur between FBN1 molecules (black and grey) that have already undergone C-terminal multimerisation and head-to-tail alignment. Compared to the wild type case (i), cMFS-associated mutations in the neonatal region (ii; orange) appear to incorporate into microfibrils at near-normal levels. In contrast, mutations affecting the neonatal region that lead to nMFS (iii; red) are unlikely to affect N- to C-terminal interactions but would disrupt later stages of assembly such as lateral association. Mutants such as ΔcbEGF16 may not directly affect the neonatal assembly site, but would affect the position and register of the region in the maturing microfibril. In addition to preventing the incorporation of mutant molecules into microfibrils, the interaction between wild type and mutant variants at step 3 would reduce the amount of wild type FBN1 deposited into the matrix, resulting in microfibril levels that are less than expected for haploinsufficiency.

platform for ligand, and contain additional post-translational modifications involved in protein recognition [62]. Thus, although all disease-causing variants with single amino acid substitutions studied here are predicted to have structural effects, only those associated with nMFS disrupt an interaction site sufficient to stall later stages of assembly.

In a previous study of cbEGF domain substitutions associated with nMFS and cMFS [34], using a mini-gene construct encoding FBN1 domains cbEGF11-22 and the same fibroblast secretion assay described here, the three calcium-binding site variants D1113G, D1115G and N1131Y variants all failed to secrete. The fold of domain cbEGF12 in this context was found to be dependent on $Ca^{2+}$ binding by cbEGF13, suggesting a post-translational rather than co-translational model of folding in this region, since cbEGF13 would be translated after cbEGF12. In addition, data from limited proteolysis experiments exposed long-range, N-terminally directed effects, with the fold of cbEGF11 being influenced by substitutions in

cbEGF13. Interactions with N-terminal domains, especially in the context of TB-cbEGF pairs, where extensive hydrophobic interfaces can be formed [4, 33], are known to stabilise the $Ca^{2+}$ binding site and therefore the fold of cbEGF domains [63]. Our new data, showing secretion of the D1113G, D1115G and N1131Y variants in the context of both the NPro-TB3cbEGF19 fusion and full-length GFP-FBN1 constructs, suggest that domain TB3 plays a stabilising role in the fold of domains cbEGF11-13 and highlight the importance of assessing, where possible, the effects of substitutions in the context of a full-length molecule. Through its interdomain interactions with cbEGF11, domain TB3 could indirectly influence the fold of cbEGF12 and may act as an intramolecular chaperone to mitigate the disruptive effects of substitutions in cbEGF13, at least to the extent that they are not detected by cell surveillance methods.

What interactions might drive higher order assembly beyond the initial C-terminal multi-merisation step? In a previous comparison of nMFS and cMFS mutants [17], FBN1 fragments harbouring nMFS substitutions were shown to have a reduced affinity for heparin compared to cMFS variants, which bound at levels similar to wild-type. The fragments used in that study, spanning from EGF4-cbEGF22, contain the entire neonatal region with just a single one of the seven known FBN1 heparin-binding sites, localised to cbEGF12-14 [64]. These data indicate that the surface properties of the EGF4 to cbEGF22 region are sufficiently altered in nMFS, but not cMFS, FBN1 variants, to affect a heparin interaction, although do not exclude a disrupted protein-protein interaction. Furthermore, our study of the ΔcbEGF16 nMFS variant shows that it is not only the cbEGF11-13 region that is critical for higher order assembly. The lack of microfibril incorporation associated with this variant indicates there are also spatial constraints on assembly in this region, since the cbEGF11-13 region is intact.

Given that 8 molecules of fibrillin are predicted to form the fibrillar structure (based on STEM mass mapping [18] and TEM imaging [65]) and the interbead region comprises near linear $Ca^{2+}$-stabilised sections, it is likely that a number of lateral interactions as well as N- to C-terminal interactions, are required to guide higher order assembly. This would explain the requirement for site-specific and length constraints. It is interesting that high resolution structural analysis of an EGF domain-rich Notch receptor/ligand complex indicates a number of primary and secondary interaction sites along a longitudinal axis, which define the orientation of the ligand relative to the receptor [66]. It is possible that fibrillin monomers are arranged in a similar way within the microfibril, with multiple interaction sites contributing to macromolecular architecture.

The involvement of the neonatal region in microfibril assembly has important implications for models of microfibril organisation. Currently, two main models exist to explain the arrangement of fibrillin monomers within microfibrils: a "pleated" model in which each fibrillin molecule spans a single interbead distance [24, 67] and a staggered model in which each monomer stretches over two interbead distances [25, 33, 37, 68]. Both models involve a head-to-tail alignment of monomers, however the staggered arrangement is dependent on at least one additional lateral association occurring to generate the observed 56 nm periodicity and would be consistent with our proposed model for the role of the neonatal region in assembly. Although the neonatal region is not currently known to interact with other FBN1 domains, it may be that earlier assembly steps (e.g., head-to-tail alignment of monomers) are required before an interaction with the neonatal region can take place, either by increasing the avidity of an interaction or through the creation of a new interacting surface.

Previously, using a similar assay, we demonstrated MFS-associated substitutions in domains TB4 and TB5 of FBN1 resulted in intracellular retention of the mutant molecules [16] whilst apparently similar mutations associated with SSS, acromicric dysplasia and geleophysic dysplasia secreted normally. In these cases, the MFS phenotype was caused by disruption of the globular structure (with a core consisting of an aromatic residue surrounded by a

conserved tetrad of salt-bridges [4, 69]), which was sufficient to result in cellular retention of the mutant form and a haploinsufficiency mechanism. SSS, acromicric dysplasia and geleophysic dysplasia variants in contrast did not alter the incorporation into microfibrils, suggesting that the disease arises due to a specific disruption of TB4/TB5 implicated in integrin-mediated signalling.

In this study we have shown that the cMFS variants are able to incorporate into 10–12 nm microfibrils in cooperation with WT protein, but the microfibril formed is likely to be somewhat defective in structure and function. In contrast, the nMFS variants are unable to assemble to form 10–12 nm microfibrils but would still be expected to have the capacity to multimerise via C-terminal sequences and form complexes with WT protein and thus create non-productive multimers. Since a threshold level of fibrillin has been shown to be important for manifestations associated with MFS, it is plausible to explain the severe effects of nMFS by a substantial reduction of microfibrils below the haploinsufficiency level, caused by a dominant negative effect. For the cMFS variants studied by this assay, we can observe incorporation of the mutant form, but we cannot exclude effects on the efficiency of assembly. Indeed, it would be unlikely for there to be no impact of structural mutations occurring at a key interaction site. Therefore, there may also be quantitative effects on assembly, as well as defective properties of microfibrils themselves.

Subtle details explaining the spectrum of milder cases of MFS are still to be determined. The R2726W substitution has been reported in several cases of mild MFS with isolated skeletal symptoms [70–72]. In COS-1 cells, this substitution reduces, but does not abolish, cleavage of the C-terminal propeptide [73], suggesting that FBN1 levels in these cases are higher than the threshold required for a complete MFS phenotype, and that a kinetic defect underlies mild disease. Allelic variation in the expression level of the wild-type allele has also been linked to intrafamilial differences in severity, with higher expression levels of the wild-type protein being associated with milder disease [13]. This effect of increased wild-type expression was clearly shown using the *Fbn1*$^{C1039G/+}$ mouse model, where introduction of a wild type human *FBN1* transgene rescued the aortic phenotype associated with the C1039G/+ background [61].

## Conclusions

This cellular study has, for the first time, distinguished differences in the ability of nMFS and cMFS variants in the TB3-cbEGF18 region of FBN1 to assemble into 10–12 nm microfibrils. Despite all the variants having structural effects on the pairwise interactions and fold of the cbEGF domains, only those associated with nMFS failed to incorporate into microfibrils. We therefore propose that nMFS arises predominantly as a consequence of dominant negative effects of mutant variants on assembly, leading to a gross reduction in microfibril quantity. cMFS by contrast either arises by a haploinsufficiency mechanism or by dominant negative effects that do not substantially affect assembly, but interfere with the integral properties of microfibrils.

## Supporting information

**S1 Table. List of all nMFS mutations in FBN1 reported in the Universal Mutation Database (http://www.umb.be/fbn1).**
(XLSX)

**S2 Table. List of cMFS and other non-nMFS mutations found in the FBN1 domains TB3-cbEGF19 (neonatal region).**
(XLSX)

**S1 Raw images.**
(PDF)

## Author Contributions

**Conceptualization:** Sacha A. Jensen, Penny A. Handford.

**Data curation:** Sacha A. Jensen, Ondine Atwa.

**Funding acquisition:** Sacha A. Jensen, Penny A. Handford.

**Investigation:** Sacha A. Jensen, Ondine Atwa, Penny A. Handford.

**Project administration:** Penny A. Handford.

**Supervision:** Sacha A. Jensen, Penny A. Handford.

**Writing – original draft:** Sacha A. Jensen.

**Writing – review & editing:** Sacha A. Jensen, Penny A. Handford.

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
