## [Decision Letter · Decision Letter 0]

24 Nov 2020

PONE-D-20-34409

Assembly assay identifies a critical region of human fibrillin-1 required for 10 - 12 nm diameter microfibril biogenesis

PLOS ONE

Dear Dr. Sacha Jensen,

Thank you for submitting your manuscript to PLOS ONE. First of all my personal apologies in the delay in getting back to you. Several reviews declined the invitation. Given that the single review is critic but positive and constructive, we invite you to submit a revised version of the manuscript that addresses all the indicatec points 

We look forward to receiving your revised manuscript.

Kind regards,

Maria Gasset, Ph.D.

Academic Editor

PLOS ONE

Journal Requirements:

"SAJ and PAH gratefully acknowledge support from Arthritis Research UK (project grant no. 20785).

www.versusarthritis.org

The funders did not play a role in the study design, data collection, data analysis, decision to publish or preparation of the manuscript. ".

i) Please provide an amended statement that declares *all* the funding or sources of support (whether external or internal to your organization) received during this study, as detailed online in our guide for authors at http://journals.plos.org/plosone/s/submit-now.  Please also include the statement “There was no additional external funding received for this study.” in your updated Funding Statement.

ii) Please include your amended Funding Statement within your cover letter. We will change the online submission form on your behalf.

Reviewers' comments:

Reviewer's Responses to Questions

**Comments to the Author**

1. Is the manuscript technically sound, and do the data support the conclusions?

Reviewer #1: Yes

2. Has the statistical analysis been performed appropriately and rigorously? 

Reviewer #1: N/A

3. Have the authors made all data underlying the findings in their manuscript fully available?

Reviewer #1: Yes

4. Is the manuscript presented in an intelligible fashion and written in standard English?

Reviewer #1: Yes

5. Review Comments to the Author

Reviewer #1: The manuscript by Jensen et al. describes important functional differences how mutations in fibrillin-1 lead to the classical Marfan syndrome and the more severe neonatal Marfan syndrome. The authors tested a series of mutations in both groups using a mini-fibrillin-1 construct transfected in the fibroblastic cell line MSU-1.1 to analyze secretion. The same mutations were introduced in full-length GFP-tagged constructs and expressed in 293 cells. These cells secrete recombinant fibrillin-1, but do not assemble them. Therefore, the authors use a previously established co-culture system with human dermal fibroblasts to test extracellular assembly of the mutant fibrillin-1 constructs. Most cMFS and nMFS mutations were secreted into the culture medium. While all cMFS mutations assembled into typical microfibrils, the nMFS mutations consistently did not assemble. This is novel and exciting, because it defines a new region (the neonatal region) as a critical determinant in microfibril assembly. The experiments are convincing and well controlled. I have the following comments.

Major comments

In the model (Fig. 5), it is a bit confusing what the authors mean with lateral assembly. I think they mean the lateral interactions between the ~8 molecules within one single microfibril. But please keep in mind that there are also lateral interactions described in the literature between individual microfibrils to produce bundles of microfibrils. The confusion stems from Fig. 5Biii where a second microfibril is indicated in the last step that is blocked by nMFS mutations. It gives the impression that the lateral bundling between microfibrils is affected. But I think this is not what the authors mean.

Also in A, step 4, it does not become entirely clear from the schematic how the green neonatal region becomes positioned at the bead region. A better explanation and color coded individual fibrillin-1 molecules would help to better understand the proposed model.

It was shown in the literature (PMID 21784848) that neonatal mutations affect the ability of mutant fibrillin-1 constructs to interact with heparin (and presumably with heparan sulfate). However, this is not well incorporated into the model. In A, the heparan sulfate chains between #3 and #4 do not interact with these moieties.

Is there any support from the literature or from the authors’ experiments that the neonatal region can physically interact with the bead region?

Since there are several fibrillin-1 assembly models available, it would be useful to discuss how the data fit (or not) with other assembly models, ideally trying to reconcile the various models.

So overall, the model requires significantly more work and refining to incorporate the existing literature and to provide a clearer representation.

Minor comments

The authors use Fbn1 as an abbreviation for human fibrillin-1 protein. As per accepted gene and protein nomenclature, human protein should be abbreviated FBN1.

6. PLOS authors have the option to publish the peer review history of their article (what does this mean?). If published, this will include your full peer review and any attached files.

Reviewer #1: No

---

## [Author Response · Author response to Decision Letter 0]

18 Jan 2021

Major comments

In the model (Fig. 5), it is a bit confusing what the authors mean with lateral assembly. I think they mean the lateral interactions between the ~8 molecules within one single microfibril. But please keep in mind that there are also lateral interactions described in the literature between individual microfibrils to produce bundles of microfibrils. The confusion stems from Fig. 5Biii where a second microfibril is indicated in the last step that is blocked by nMFS mutations. It gives the impression that the lateral bundling between microfibrils is affected. But I think this is not what the authors mean.

Also in A, step 4, it does not become entirely clear from the schematic how the green neonatal region becomes positioned at the bead region. A better explanation and color coded individual fibrillin-1 molecules would help to better understand the proposed model.

- Figure 5 has now been updated to take into account the reviewer's comments. In panel B, where we depict the lateral assembly that we propose is driven by the neonatal region, we have provided more details on the final assembly step (4) shown in panel A. We also now show a second set of fibrillin monomers in grey, aligned head-to-tail, and how these microfibril precursors may interact laterally to produce the mature structure.

It was shown in the literature (PMID 21784848) that neonatal mutations affect the ability of mutant fibrillin-1 constructs to interact with heparin (and presumably with heparan sulfate). However, this is not well incorporated into the model. In A, the heparan sulfate chains between #3 and #4 do not interact with these moieties.

- Panel A of figure 5 has been simplified to reflect the interactions between heparan sulphate and specific sites on fibrillin-1. Only 3 of the 7 known heparan interaction sites, at the N- and C- termini and the neonatal region, have been shown for clarity. 

Is there any support from the literature or from the authors’ experiments that the neonatal region can physically interact with the bead region?

- The neonatal region of fibrillin-1 is not currently known to interact with any other fibrillin-1 domains. It may be that earlier assembly steps are needed before the required spatial arrangement of domains is created in the nascent microfibril to allow an interaction with the neonatal region. We have therefore added a comment in the discussion (p.20, line 4) to address this.

Since there are several fibrillin-1 assembly models available, it would be useful to discuss how the data fit (or not) with other assembly models, ideally trying to reconcile the various models.

- We have now included a paragraph that addresses this point at the end of p.19 of the discussion, describing how our model for the role of the neonatal region in assembly is more compatible with a staggered model of microfibril organisation.

Minor comments

The authors use Fbn1 as an abbreviation for human fibrillin-1 protein. As per accepted gene and protein nomenclature, human protein should be abbreviated FBN1.

- This has now been corrected.

---

## [Decision Letter · Decision Letter 1]

19 Feb 2021

PONE-D-20-34409R1

Assembly assay identifies a critical region of human fibrillin-1 required for 10 - 12 nm diameter microfibril biogenesis

PLOS ONE

Dear Dr. Sacha Jensen,

Thank you for submitting your revised manuscript to PLOS ONE and include all indicated amendements. However before its acceptance a minor correction must be performed. Define fibrillin-1 abbreviation as FBN1 after the first time fibrillin-1 is mentioned and substitute the full name along the rest of the text and in figures (plots and legends).

We look forward to receiving your revised manuscript.

Kind regards,

Maria Gasset, Ph.D.

Academic Editor

PLOS ONE

Reviewers' comments:

Reviewer's Responses to Questions

**Comments to the Author**

1. If the authors have adequately addressed your comments raised in a previous round of review and you feel that this manuscript is now acceptable for publication, you may indicate that here to bypass the “Comments to the Author” section, enter your conflict of interest statement in the “Confidential to Editor” section, and submit your "Accept" recommendation.

Reviewer #1: All comments have been addressed

2. Is the manuscript technically sound, and do the data support the conclusions?

Reviewer #1: Yes

3. Has the statistical analysis been performed appropriately and rigorously? 

Reviewer #1: N/A

4. Have the authors made all data underlying the findings in their manuscript fully available?

Reviewer #1: Yes

5. Is the manuscript presented in an intelligible fashion and written in standard English?

Reviewer #1: Yes

6. Review Comments to the Author

Reviewer #1: All critiques have been appropriately addressed.

7. PLOS authors have the option to publish the peer review history of their article (what does this mean?). If published, this will include your full peer review and any attached files.

Reviewer #1: No

---

## [Author Response · Author response to Decision Letter 1]

25 Feb 2021

Dear Editor,

 Thank you for the recent response to our manuscript, “Assembly assay identifies a critical region of human fibrillin-1 required for 10 - 12 nm diameter microfibril biogenesis” (manuscript number: PONE-D-20-34409R1). We were asked to make a minor correction to the manuscript before its acceptance, which was to define the fibrillin-1 abbreviation as FBN1 after the first time fibrillin-1 is mentioned and to then substitute the full name through the rest of the article (including figures and legends). This has now been done, with the FBN1 abbreviation being defined at its first mention in the abstract. 

 A few other minor typographical corrections have also been made, including the renaming of the recombinant GFP-tagged version of fibrillin-1 from "GFP-FBN" to the more specific "GFP-FBN1". In addition, after checking the figures using the PACE digital diagnostic tool, it was noted that figure 4 did not meet PLOS requirements and has therefore been divided into two new figures. All changes are highlighted in the marked-up version of the manuscript.

 We would also like to make a change to our financial disclosure statement to acknowledge funding from the UK Medical Research Council (MRC), which contributed to the work in this manuscript. The updated statement would read:

SAJ and PAH gratefully acknowledge support from Arthritis Research UK (www.versusarthritis.org; project grant no. 20785) and the MRC (https://mrc.ukri.org; grant number MR/M009831/1).

Yours sincerely, 

Sacha Jensen

25/2/21

---

## [Editor Report · Decision Letter 2]

1 Mar 2021

Assembly assay identifies a critical region of human fibrillin-1 required for 10 - 12 nm diameter microfibril biogenesis

PONE-D-20-34409R2

Dear Dr. Sacha Jensen,

We’re pleased to inform you that your manuscript has been judged scientifically suitable for publication and will be formally accepted for publication once it meets all outstanding technical requirements.

Kind regards,

Maria Gasset, Ph.D.

Academic Editor

PLOS ONE
---

## [Editor Report · Acceptance letter]

10 Mar 2021

PONE-D-20-34409R2 

Assembly assay identifies a critical region of human fibrillin-1 required for 10 - 12 nm diameter microfibril biogenesis 

Dear Dr. Jensen:

I'm pleased to inform you that your manuscript has been deemed suitable for publication in PLOS ONE. Congratulations! Your manuscript is now with our production department. 

Kind regards, 

on behalf of

Dr. Maria Gasset 

Academic Editor

PLOS ONE